# Stakeholder Perspectives of Australia’s National HPV Vaccination Program

**DOI:** 10.3390/vaccines10111976

**Published:** 2022-11-21

**Authors:** Caitlin Swift, Aditi Dey, Harunor Rashid, Katrina Clark, Ramesh Manocha, Julia Brotherton, Frank Beard

**Affiliations:** 1National Centre for Immunisation Research and Surveillance, Sydney Children’s Hospitals Network, Sydney, NSW 2145, Australia; 2Faculty of Medicine and Health, The University of Sydney, Sydney, NSW 2006, Australia; 3Healthed, Sydney, NSW 1805, Australia; 4Australian Centre for the Prevention of Cervical Cancer, Melbourne, VIC 3053, Australia; 5Melbourne School of Population and Global Health, University of Melbourne, Melbourne, VIC 3010, Australia

**Keywords:** human papillomavirus (HPV), HPV vaccine, vaccination program, evaluation, stakeholder, survey, interviews

## Abstract

Australia has been a world leader in human papillomavirus (HPV) vaccination and was the first country to implement a fully funded national HPV vaccination program, from 2007 for girls and 2013 for boys. In 2018 the program changed from a 4-valent to 9-valent HPV vaccine and a 3-dose to 2-dose standard schedule. We assessed stakeholder perspectives on factors influencing program outcomes and impact as part of a comprehensive program evaluation. In late 2019 and early 2020, we conducted 26 interviews with 42 key stakeholder participants and received 1513 survey responses from stakeholders including general practice staff and school-based nurse immunisers. Findings included that the 2-dose schedule is better accepted by schools and students and has reduced program cost and resource requirements. However, course completion rates have not increased as much as anticipated due to the 6–12 month dosing interval and reduced opportunities for school-based catch-up vaccination. Major reported barriers to increased vaccine coverage were absenteeism and consent form return. Vaccine hesitancy is not currently a major issue but remains a potential threat to the program. While Australia’s HPV vaccination program is perceived as highly successful, measures to further enhance the program’s impact and mitigate potential threats are important.

## 1. Introduction

Human papillomavirus (HPV) is a common sexually transmitted infection with potential to cause both malignant and non-malignant disease. Oncogenic HPV types (particularly HPV16/18) can cause cervical cancer, other anogenital cancers (such as anal, penile, vaginal and vulval cancer) and oropharyngeal cancers [1]. Non-oncogenic HPV types can cause genital warts [2] and recurrent respiratory papillomatosis [3].

Cervical cancer is the fourth most common cancer and fourth most frequent cause of cancer mortality in females worldwide [4], with a higher incidence and mortality seen in lower-resourced countries compared to higher-resourced countries [5]. To address this high burden of disease and global inequity, the World Health Organization (WHO) has developed a global strategy for the elimination of cervical cancer as a public health problem [6]. As part of the strategy, all countries are recommended to achieve the ‘90–70–90′ targets by 2030: (1) 90% of girls fully vaccinated with HPV vaccine by age 15 years; (2) 70% of women screened with a high-performance test by 35 years of age and again at 45 years of age; and (3) 90% of women identified with cervical disease receive treatment [6].

Australia’s National Cervical Cancer Elimination Strategy (under development) will aim to eliminate cervical cancer as a public health problem by 2035 [7]. Modelling suggests Australia may achieve this at the overall population level as early as 2028 [8], but elimination is not on track for Aboriginal and Torres Strait Islander women [9]. Addressing inequities within Australia’s cervical cancer control strategies, including HPV vaccination, is necessary to ensure cervical cancer elimination across all population subgroups [10].

Importantly, HPV vaccination also has other benefits for both sexes including observed reductions in genital warts [11,12,13] and juvenile-onset recurrent respiratory papillomatosis [14,15], and potential to prevent other anogenital cancers [16,17,18] and oral cancers [19].

Australia has been a world leader in HPV vaccination as the first country to implement a National HPV Vaccination Program, added in 2007 to the National Immunisation Program, which is run collaboratively by the Australian federal government and eight state and territory governments [20]. The program commenced with a 3-dose schedule of the quadrivalent HPV (4vHPV) vaccine Gardasil^®^ delivered through schools for girls aged 12–13 years, with a catch-up program available to end 2009 for females aged up to 26 years [21]. The program was expanded to include boys aged 12–13 years in 2013, with catch-up vaccination available until end 2014 for boys aged 14–15 years [21].

In 2018, the standard 3-dose schedule of 4vHPV vaccine was replaced with a 2-dose schedule of the 9-valent HPV (9vHPV) vaccine Gardasil9^®^. One state (New South Wales [NSW]) made this change in 2017. This reduction to two doses was accompanied by expectations of greater acceptability, increased coverage and greater reductions in HPV-related disease [22]. Australian adolescents aged <15 years are now recommended to receive two doses of 9vHPV vaccine 6–12 months apart, while adolescents aged ≥15 years or with significant immunocompromise are recommended to receive three doses of 9vHPV vaccine at 0, 2 and 6 months [23]. Australia’s HPV vaccination program continues to be primarily school-based but two doses of community-based catch-up vaccine are funded up to age 19 years [24].

A whole-of-life Australian Immunisation Register (AIR) was established in late 2016 and all HPV vaccination records were transferred from the previous National HPV Vaccination Program Register to the AIR in 2018 [25]. All HPV vaccinations given in Australia must now be reported directly to the AIR.

The first evaluation of Australia’s National HPV Vaccination Program published in 2014 focused on implementation of the program, including a process evaluation [26]. We assessed stakeholder perspectives on impacts of the program since the previous evaluation, as part of a larger impact evaluation, with the aim of identifying factors that have positively or negatively influenced program outcomes and impacts, or could do so, with particular emphasis on vaccination coverage and strategies to increase coverage and address inequities.

## 2. Materials and Methods

We used a mixed-methods approach to collect both qualitative and quantitative data through interviews with key stakeholders and an online survey of other stakeholders across Australia, predominantly immunisation providers. Semi-structured telephone interviews were conducted with key stakeholders from October 2019 to March 2020. Participants were recruited directly by purposive sampling based on their role in the program, area of expertise or on referral by other key stakeholders. An interview questionnaire was developed based on the previous evaluation [26], covering HPV vaccination coverage, vaccine safety, reporting to the AIR, interrelationship between the HPV vaccination and cervical screening programs, strengths and challenges of the HPV vaccination program and recommendations for enhancing impact. The questionnaire was tailored for each stakeholder group to ensure the topic areas were relevant to their work or expertise, and emailed to participants ahead of their interview. Interview responses were thematically analysed, and quotes selected that illustrate key findings.

An anonymous online SurveyMonkey^®^ survey was developed covering the same topics as the key stakeholder interviews. The survey weblink was emailed to school-based immunisation providers in NSW, Australian Capital Territory (ACT), Northern Territory (NT), Queensland, South Australia (SA), Victoria and Western Australia (WA) via state and territory immunisation program managers and other pathways suggested by them. The survey was not distributed to school-based immunisation providers in Tasmania due to separate qualitative research being undertaken concurrently. The survey weblink was also emailed to other immunisation providers in collaboration with Healthed, a national provider of health education for the primary care sector, and distributed to Aboriginal Community Controlled Health Service (ACCHS) staff in NT, SA and Tasmania. The survey was open for completion at different times for the various stakeholder groups from November 2019 to February 2020. A descriptive analysis was conducted of survey response frequencies for each question and thematic analysis conducted of free-text comments, with quotes selected to illustrate key findings.

An Advisory Committee and Cultural Advisory Committee provided oversight of the study. Ethics approval was obtained from Sydney Children’s Hospitals Network Human Research Ethics Committee (HREC), the NT Health and Menzies School of Health Research HREC, the Aboriginal Health and Medical Research Council of NSW HREC, and the Aboriginal HREC of SA.

## 3. Results

### 3.1. Key Stakeholder Interviews

A total of 26 telephone interviews were conducted with 42 key stakeholder participants (Table 1).

#### 3.1.1. Experience of 2 Dose Transition

Key stakeholders identified many benefits of Australia’s transition to a two-dose schedule of 9vHPV vaccine. These included the greater ease of scheduling school visits, greater acceptance by student and parents due to only needing two doses/needles, and reductions in required program resources, costs, and data collection. Representatives from all states and territories reported using a dosing interval of 6–7 months in schools, with dose 1 delivered in the first half of the school year and dose 2 in the second half. Key stakeholders had generally expected that moving to two doses would lead to an increase in course completion rates, but six of eight state and territory immunisation managers were concerned that this had not been realised or was less than expected in their jurisdiction. This was attributed to dose 2 being delivered later in the year when student absenteeism is higher, school scheduling is more difficult due to study and exam periods, and there are reduced opportunities for school-based catch-up vaccination. Retaining a casual nursing workforce was also noted to be difficult in some areas due to the longer dosing interval compared to the 3-dose schedule.

#### 3.1.2. Consent Forms

Student absenteeism and consent form return were the two major factors identified by key stakeholders as negatively influencing the impact of the school-based vaccination program. Two states were reported to be developing electronic consent forms at the time interviews were conducted, aiming to remove reliance on students taking home and returning paper forms. Immunisation staff from all but one of the other states and territories also expressed interest in developing electronic consent forms. However, difficulties related to use of electronic consent and other digital technology in remote areas were identified, along with the need for additional training of staff.

Other initiatives to improve consent form return implemented by states and territories since the previous evaluation included: actively following up unreturned forms by phone call, email, letter or SMS to parents; putting ‘sign consent forms’ at the top of the ‘to-do’ list in school packs for parents; HPV vaccine promotion at the end of the year before vaccination is due so students and parents know to expect consent forms early the following year; verbal parental consent obtained on vaccination day; addressing common parental concerns on the consent form; and allowing mature minors to consent for themselves with school principal endorsement. Differences across states and territories were apparent, for example immunisation staff in one state reported they could not access parent contact details until a consent form was returned, which meant they were unable to follow up unreturned forms; while staff from two other states reported that their public health legislation had been amended so that information could be obtained from schools for this purpose.

#### 3.1.3. Delivery Models

The interviews also revealed considerable differences in how the school-based immunisation program is operationalised by states and territories, which can impact the capacity for initiatives to improve HPV vaccination coverage. The program may be delivered by state and territory health department staff (including public health units, school-based nurses or remote area nurses depending on the location), local council immunisation staff, or subcontracted to third party immunisation providers, with variation both within and across states and territories. Identified issues with these models included budget cuts within local councils reducing capacity to follow up unreturned forms and absent students, and perceived disinvestment by third party immunisation providers leading to reduced performance. Differences in capacity for school-based catch-up vaccination across states and territories were also identified.

#### 3.1.4. Strategies for Aboriginal and Torres Strait Islander Students

For Aboriginal and Torres Strait Islander populations, effective strategies to achieve high HPV vaccination coverage were reported as: having highly engaged school staff; utilising school Aboriginal Liaison Officers; speaking directly with adolescents and families about HPV vaccination; partnering with community organisations and Aboriginal Medical Services; having the support of community Elders; incentives like T-shirts; and recall/reminder systems.


*“Health checks, you can get a T-shirt and … a gift voucher. But we’re not funded for that stuff and the best way, for Australia and getting their immunisations rate up, would be to offer an incentive.”*
(ACCHS Nurse Immuniser)

Language barriers amongst Aboriginal and Torres Strait Islander peoples were noted in some areas but could be overcome with family or clinic staff acting as interpreters. Information and consent materials with culturally appropriate language were deemed necessary, with consideration also needed regarding the best means of providing information (e.g., online information on a tablet device versus written resources). The recognition of separate men’s and women’s health issues in relation to HPV vaccination when providing information was also considered important. To help facilitate parental consent, the NT had developed a culturally appropriate simplified consent form for Aboriginal adolescents from remote communities who attend boarding schools.


*“Signing a consent document which is very wordy and in English, can be a barrier for some students to get their parents to sign or to have that full understanding of what … vaccines their children are having.”*
(Immunisation Public Health Nurse)

High rates of student absenteeism, and being a more mobile population with frequent changes in address and phone numbers, were other reported challenges to HPV vaccination amongst Aboriginal and Torres Strait Islander populations. It was reported to be easier to address these issues in smaller communities where staff know everyone than in larger metropolitan areas. Parental safety concerns were also perceived as having a negative impact in some Aboriginal and Torres Strait Islander communities, including the misconception that the HPV vaccine can cause cervical cancer.

Initiatives planned by state and territory health departments to increase coverage in Aboriginal and Torres Strait Islander adolescents included: funding dedicated nurse immunisers for Aboriginal children and adolescents; extending school-based catch-up vaccination for Aboriginal and Torres Strait Islander students up to and including the final year of school; and offering targeted catch-up vaccination clinics in the second half of the school year for schools with high proportions of Aboriginal and Torres Strait Islander students.

#### 3.1.5. Strategies for Culturally and Linguistically Diverse Students

For culturally and linguistically diverse (CALD) populations, translated resources were available through the Australian Government Department of Health and had also been developed independently by some states and territories and local immunisation providers. Seqirus had also developed translated resources on 9vHPV vaccine and HPV disease awareness for people attending general practice (GP) clinics. A challenge identified was that access to interpreters was not funded for school-based immunisers. Refugees were noted to be very compliant with vaccinations but a lack of language support for refugees with less common languages was reported as challenging, with reliance on family to interpret. Other reported challenges were identifying adolescents from CALD communities and insufficient data to identify gaps in vaccination coverage among CALD communities, to inform any initiatives required.


*“We do have CALD resources in 15 different languages, but apart from that we don’t have specific program things, I don’t think. And also we don’t have any research about that particular community, how well they are accessing the HPV vaccine. Because it’s obviously not recorded on the AIR, it’s hard to get that data.”*
(State Immunisation Program Manager)

#### 3.1.6. Strategies for Students with Lower Socioeconomic Status

For populations of low socioeconomic status (SES), ensuring free access to vaccination was considered a priority, through either a school-based catch-up clinic, local council clinic or bulk-billing GP (i.e., a GP who directly bills Medicare, the Australian universal health insurance scheme, with no additional charge to patients), with a lack of bulk-billing GPs in some areas a challenge. Similar to CALD populations, difficulty identifying adolescents from low SES backgrounds was a reported challenge, with a reliance on identification by school staff.

#### 3.1.7. Vaccine Hesitancy

Key stakeholders perceived that HPV vaccine hesitancy was not significantly impacting HPV vaccination coverage in Australia overall, with impacts limited to specific areas and groups. The influence of social media on HPV vaccination coverage was perceived as both positive, through the opportunity to promote the vaccination program, and negative, through spreading of anti-vaccination views, although these were deemed to gain little traction within the broader population. Monitoring of community sentiment and social media messaging was considered important given the potential for rapid spread of misinformation.

#### 3.1.8. Vaccine Coverage Target

Key stakeholders, including the Australian Government Department of Health immunisation staff and seven of eight state or territory immunisation managers, thought that the WHO target of 90% 2-dose vaccination coverage for females by 15 years was achievable in Australia with concerted efforts, including raising awareness, better reporting to the AIR and identification of coverage gaps and reasons for them. The achievement of vaccination rates above 90% for childhood vaccines, and for the first dose of HPV vaccine in many areas, were cited as evidence of Australia’s potential to achieve this target.

### 3.2. Online Survey

A total of 1513 people responded to the online survey, with characteristics summarised in Table 2.

#### 3.2.1. Experience of 2 Dose Transition

Similarly to the key stakeholder interviews, survey respondents largely identified advantages of the 9vHPV vaccination program as the ease, acceptability and reduced resourcing related to the 2-dose schedule. The increased valency of the 9vHPV vaccine was also recognised as having good community support and helping with promotion of the vaccine. Many respondents thought that the change to 9vHPV vaccine had increased coverage for adolescents aged <15 years in their area, particularly for dose 2 (Figure 1).


*“Schools are happy that we only need two visits to complete the series for the year. It is also easier to staff for two visits instead of three and costs less in wages for staff.”*
(Regional area local council nurse immuniser)

Most respondents (837, 74.2%) reported no issues with implementing the 9vHPV vaccination program. Issues were reported by 149 (13.2%) respondents, including safety rumours, queries about whether revaccination with 9vHPV vaccine is required after receiving 4vHPV vaccine, confusion about eligibility and determining the number of 9vHPV doses required, and issues with payment for the recommended third dose when it is not funded under the National Immunisation Program.

#### 3.2.2. School-Based Vaccination Coverage

As with the key stakeholder interviews, survey respondents identified consent form return and student absenteeism as the factors most frequently impacting school-based HPV vaccination coverage in their area (Figure 2).

A total of 849 survey respondents provided additional comments about strategies to improve school-based HPV vaccination coverage in their area. The most frequent of these, suggested in half the comments, was increased education for parents and adolescents (females and males), including regarding the benefits of HPV vaccination. Enhanced use of digital/electronic technology and the use of reminders were each suggested in a quarter of comments, although a few noted that digital technology would not be suitable for all parents, such as those without email addresses or whose phone numbers change regularly. Other comments suggested a need for better support and recognition of the importance of the program from some schools.


*“Electronic consent forms would circumvent students not passing consent forms on to their parents and/or not returning them to the school. It would also eliminate the middle man (i.e., the school/teachers) & avoid issues such as failure to give consent forms to all students … and misplaced/lost consent forms.”*
(Regional area school-based nurse immuniser and practice nurse)


*“Digital reminders and increased education sessions of both the students and parents prior to the vaccination. An info brochure is not enough, as many students don’t know what they are being vaccinated for at the time and don’t remember later on.”*
(Remote area women’s health nurse and immuniser)

#### 3.2.3. Role of Primary Care

One-third of survey respondents (375/1128) perceived HPV catch-up vaccination in primary health care settings to be under-utilised for adolescents aged ≥15 years, and one-quarter (276/1128) for adolescents aged <15 years. Approximately half the respondents (548/1128) were satisfied with the relationship between the school-based HPV vaccination program and primary care providers in their area, with almost one-quarter (248/1128) not satisfied and the remainder unsure. Issues identified in respondents’ comments included: lack of public awareness of this catch-up option (in primary care); lack of communication between the school-based vaccination program and primary care; GP clinics not being aware of which adolescents have missed doses at school; GP clinics not having HPV vaccine in stock; and confusion amongst both GPs and parents about funded HPV vaccine eligibility.


*“The process to obtain vaccines is not straightforward, as GPs are not allowed to have any stock but rather have to order it in individually for each person. This creates potentially two visits to the GP, which they may choose to charge a fee for.”*
(Major city school-based nurse immuniser)


*“There is no interaction between local schools and my practice. I will not know which children have missed out on the day of vaccine in their school, unless they are self-presenting to my clinic.”*
(Major city General Practitioner)

#### 3.2.4. Priority Populations

Similarly to the overall population, survey respondents identified the major challenges to HPV vaccination in Aboriginal and Torres Strait Islander populations to be consent form return and absenteeism. Strategies reported as effective in addressing these challenges included: utilisation of Aboriginal Liaison Officers or Aboriginal Health Workers to assist with consent form return and support students on vaccination day; home visits to provide catch-up vaccination; and obtaining verbal consent from parents or carers on vaccination day.


*“Return of consent forms and attendance very poor. Many verbal consents are obtained on site. This is very time consuming due to the high number we do … Liaising with Aboriginal Health and Community Health Centres has been successful to a degree.”*
(Major city school-based nurse immuniser)

Language was identified in comments as a barrier to HPV vaccination in CALD populations, reported as difficult to identify via schools and contributing to a lack of parental understanding and failure to consent or incorrect completion of consent forms. Being able to provide education and address misconceptions was suggested as an effective way to obtain consent in CALD populations, e.g., speaking directly with parents or immunisation staff attending school information days, with the use of interpreters as required. Translated educational materials were also reported to be insufficient at times or not always available.


*“Difficulty getting information from school on the language spoken at home and so the correct consent form to give to the family. Lack of understanding on what the vaccine is for. Even when a self-addressed envelope was sent to parents from one school for easy return to us, the rate was still very low of returned cards.”*
(Regional area school-based nurse immuniser)

For people of low SES, obtaining consent and absenteeism were again highlighted as the major barriers to HPV vaccination. Opportunistic checking of adolescent vaccination histories was reported by primary care and community health staff as a strategy to catch up adolescents with missed doses. It was also noted that vaccinating adolescents being managed within government child protection systems is challenging as they move schools frequently.


*“We find that we often get lower uptake in schools in areas of socioeconomic disadvantage—less consent forms returned for the first dose as well as increased absenteeism.”*
(Regional area school-based nurse immuniser)

#### 3.2.5. Vaccine Hesitancy

Of 1090 respondents, most reported encountering vaccine hesitancy either rarely (351, 44.1%) or sometimes (341, 42.9%) in their work, with only 54 (6.8%) frequently or very frequently. Safety concerns were perceived to be more important than religious or philosophical objections in contributing to HPV vaccine hesitancy, and adolescent needle phobia or fear was also identified as contributing in comments. The reason for parents not consenting to HPV vaccination was noted to not always be documented on returned forms.

#### 3.2.6. Coverage Target

Most survey respondents (864, 84.4%) thought that the WHO vaccination target is achievable in Australia. Only 47 (4.6%) thought it is unachievable with 113 (11.0%) unsure. Common suggestions of how to achieve the target included: greater engagement from schools; enhanced funding and resources for immunisation staff to follow up students not vaccinated at school; incentives e.g., inclusion of HPV vaccine in the Australian government’s ‘No Jab No Pay’ policy; and enhanced public messaging campaigns about the benefits of HPV vaccination and impact on disease.


*“I feel there should be more responsibility on the schools/teachers to assist with the return of consent forms. The ones that have no desire to chase up consent forms have no consequences, return rates are low, vaccination rates are low and it’s almost impossible for us to chase up these students.”*
(Major city school-based nurse immuniser)

## 4. Discussion

Our study findings show that stakeholders perceive Australia’s HPV vaccination program as highly successful, but that new approaches may be required to further increase HPV vaccination coverage and maximise the impact of the program.

The many perceived benefits of the change to a standard 2-dose 9vHPV vaccination schedule include simpler logistics, greater acceptance and reduced resourcing requirements. However most key stakeholders acknowledged that the impact of this schedule change on coverage has not been as large as they had anticipated. There have been only small increases in HPV course completion since the reduction to two doses [27], with national HPV course completion rates by age 15 years reaching 80.5% for females and 77.6% for males in 2020 (75.0% and 68.0% for Aboriginal and Torres Strait Islander females and males, respectively) [28]. It has also been noted that 2-dose coverage achieved in 2020 under the new schedule was lower than in previous years when the second dose was part of a 3-dose schedule [27].

Our study identifies some of the challenges involved. In the previous 3-dose schedule, dose 1 or 3 were more likely to be missed in schools than dose 2 [29]. Stakeholders in our study reported that giving dose 2 later in the school year after at least a 6-month interval is challenging for logistical reasons including higher rates of student absenteeism, difficulty scheduling school visits due to this being a busy period for schools and reduced opportunities for school-based catch-up vaccination. These barriers will likely need to be addressed to see significant further improvements in course completion rates in Australia. Further research to specifically explore factors impacting HPV vaccination in Aboriginal and Torres Strait Islander adolescents is also required, and underway [30].

We found that like the previous evaluation [26], consent form return and student absenteeism continue to be the major perceived barriers to HPV vaccination in Australian schools. These two factors have been associated with lower vaccination coverage in Australian schools, along with low parental literacy, language barriers and difficulty contacting parents [31]. Poor parental and student knowledge regarding HPV infection, HPV vaccine and immunisation in general are also barriers [32], supporting the recommendations made by stakeholders in our study for increased education. Use of digital technology such as electronic consent forms and reminders to parents was also recommended as a strategy to increase HPV vaccination coverage, although projects to progress electronic consent were only underway in two states at the time of our study.

Inclusion in Australia’s National Immunisation Program and the school-based delivery model are key drivers of parental acceptance and decision-making about HPV vaccination in Australia [33], with high public support for school-based vaccination programs [34]. Given the challenge of student absenteeism, particularly late in the year, enhancing opportunities for school-based catch-up vaccination is likely to be important for improving HPV vaccination coverage in Australia. We identified variable capacity for school-based catch-up vaccination across Australia though, as also found in the previous evaluation [26], making this a potential area of improvement for program delivery if adequately resourced.

Our study also found that many stakeholders considered HPV catch-up vaccination through primary care to be under-utilised, with limited communication between school programs and GP clinics, and poor community awareness of this option. Enhancing communication and raising awareness of this catch-up option will be important to supplement school-based catch-up arrangements.

Regarding CALD populations, providing tailored education can improve HPV vaccination uptake amongst immigrant parents [35]. Barriers to HPV vaccine acceptability and uptake among parents of Arabic background in Australia have included a lack of knowledge and awareness of HPV and HPV vaccination, specific religious and cultural barriers (such as whether the vaccine is Halal) and a lack of translated information [36]. These issues are broadly consistent with our study findings, and support the need for greater capacity within Australian school-based vaccination programs to provide tailored support and education for people of CALD backgrounds. As suggested by key stakeholders in our study, enhanced analysis of vaccine coverage data would also help identify coverage gaps within CALD populations and facilitate development of strategies to improve uptake. While the AIR does not contain information on country of birth or ethnicity, this could be achieved through linkage of AIR data with other administrative datasets.

Stakeholders in our study generally did not perceive vaccine hesitancy to have a significant impact on HPV vaccination coverage in Australia. However, comparison of HPV and diphtheria-tetanus-pertussis vaccine coverage in Australian schools (routinely given at the same visit) shows possible pockets of HPV-specific vaccine hesitancy, with lower HPV vaccination coverage in some areas [37]. Our study identified that stakeholders consider monitoring of community perceptions and social messaging regarding HPV vaccination to be important, given the potential for rapid spread of misinformation.

Highlighting the strength of the program in Australia, most stakeholders in our study thought that the WHO target of 90% female 2-dose coverage by 2030 is achievable, with concerted effort. Coverage has been steadily increasing since the program was implemented [28,38] and the current national target is 85% for both sexes, as per the National Preventive Health Strategy 2021–2030 [39]. Sustained coverage of ≥80% in a both-sex HPV vaccination program such as Australia’s should result in local elimination of targeted HPV types [40]. Globally, 90% is an ambitious target, with 2019 estimates showing only five countries had achieved this target and many countries yet to introduce HPV vaccine [41]. Many stakeholders in our study expressed interest in a single dose strategy to help achieve a 90% target in Australia. In April 2022, the Strategic Advisory Group of Experts on Immunisation of the WHO recognised a single dose as providing comparable efficacy to 2- or 3-dose regimens [42], with an updated WHO position paper awaited.

A strength of this study was the high number of responses to the online survey, through which we obtained rich perspectives from immunisation providers who directly interact with schools, adolescents and parents, and also demonstrated the high level of stakeholder interest and support the program has in Australia. Our key stakeholder interviews were conducted with all state and territory immunisation managers or equivalent, which provided comprehensive insights into successes and challenges of the program across Australia, and identified the different strategies developed to increase coverage. Monitoring the impact of these strategies will be important.

There are several limitations of this study. The perspectives obtained may not be fully representative of relevant stakeholder groups, in particular groups with limited representation such as ACCHS staff and immunisation providers in smaller states and territories. The majority of respondents to the online survey were female and middle aged or older, which may partly reflect the demographics of the targeted stakeholders (nursing being a predominantly female profession in Australia [43]), but could also reflect greater likelihood of response among people with personal experience of HPV-related disease, although this would seem unlikely to be a significant source of bias given the nature of questions involved. Our survey respondents were diverse and included non-school-based immunisation staff, which may account for the variable proportions of responses selected as ‘Don’t know or N/A’, however the impact of this on reliability of our conclusions is likely limited. Our findings may also not have identified all factors influencing impact of the HPV vaccination program across Australia, given regional and population diversity and identified differences in program implementation. We also did not assess perspectives of school staff, parents and adolescents, who are other key stakeholders in the program and whose perspectives may differ from those reported.

As this study was completed just before the COVID-19 pandemic emerged, stakeholder perspectives may have evolved. Early findings indicated limited overall impact of the pandemic on routine vaccination coverage in Australia [44], but the proportion of adolescents completing the 2-dose HPV vaccination schedule within a calendar year was 12 percentage points lower in 2020 than in 2019, likely due to pandemic-related disruption to school-based programs [20]. With most COVID-19 restrictions now lifted in Australia, efforts to ensure high levels of catch-up vaccination and close monitoring of HPV vaccination coverage for the affected age cohorts are even more important. Our pre-pandemic study provides a baseline against which to further assess pandemic-induced impacts on the program in Australia.

## 5. Conclusions

This study provides a comprehensive assessment of stakeholder perspectives on Australia’s National HPV Vaccination Program and identifies ways the program’s impact could be enhanced. Our findings should be of interest to immunisation policymakers, program staff and researchers in other countries. Monitoring vaccine coverage and community sentiment towards HPV vaccination are crucial as we enter the post-pandemic period and continue progress towards global cervical cancer elimination.

## Figures and Tables

**Figure 1 vaccines-10-01976-f001:**
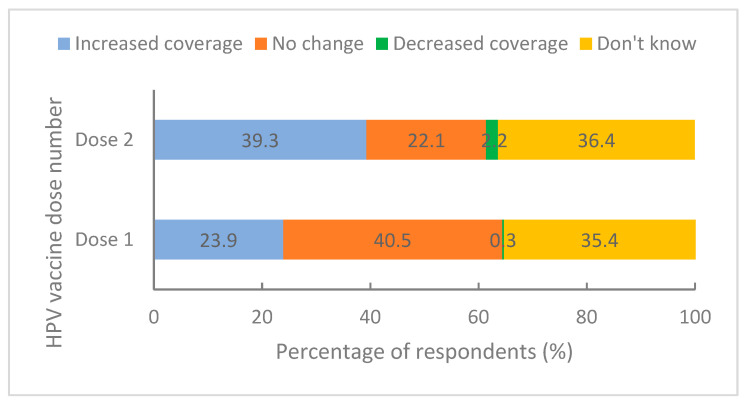
Survey respondent perceptions of the influence of the change to a 2-dose schedule on HPV vaccination coverage for adolescents aged <15 years in their area (*n* = 1128).

**Figure 2 vaccines-10-01976-f002:**
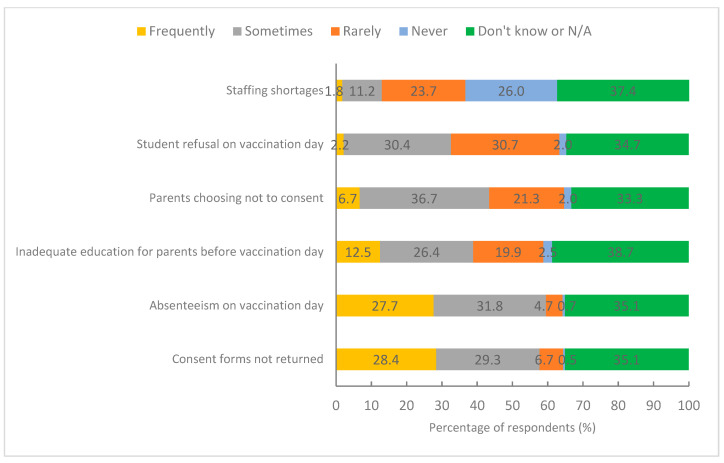
Survey respondent perception of the frequency of factors impacting school-based HPV vaccination coverage (*n* = 447).

**Table 1 vaccines-10-01976-t001:** Key stakeholder interview participants by stakeholder group.

Key Stakeholder Group	Participants *n* (%)
State and territory immunisation program managers and associated staff; and local council immunisation staff	19 (45.2)
Aboriginal Community Controlled Health Service staff and remote area immunisation coordinators	9 (21.4)
Australian Government Department of Health immunisation staff and Therapeutic Goods Administration *	5 (11.9)
Australian Government Department of Health, Cervical Screening Section and state-level cervical screening program manager	3 (7.1)
HPV vaccination researchers and sexual health physician	3 (7.1)
Seqirus †	3 (7.1)
**Total**	**42 (100)**

* Australian Government agency responsible for regulation of vaccine use (consolidated written response provided). † Distributor of HPV vaccine in Australia.

**Table 2 vaccines-10-01976-t002:** Characteristics of survey respondents.

Respondent Characteristics	Respondents *n* (%)
**Occupation**	
GP	778 (51.4)
GP practice nurse	210 (13.9)
School-based nurse immuniser	166 (11.0)
Aboriginal Health Worker	5 (0.3)
Other *	354 (23.4)
**Gender**	
Female	1319 (87.2)
Male	189 (12.5)
Other	5 (0.3)
**Age group**	
<25 years	6 (0.4)
25–34 years	123 (8.1)
35–44 years	287 (19.0)
45–54 years	416 (27.5)
55 years and over	681 (45.0)
**Location of employment**	
New South Wales	477 (31.5)
Victoria	403 (26.6)
Queensland	260 (17.2)
Western Australia	184 (12.2)
South Australia	115 (7.6)
Northern Territory	26 (1.7)
Australian Capital Territory	24 (1.6)
Tasmania	19 (1.3)
Other	14 (0.9)
**Rurality of employment**	
Major city	911 (60.2)
Regional	511 (33.8)
Remote	91 (6.0)

* Roles reported included immunisation coordinators, community-based nurses, women’s health nurses, midwives, pharmacists, gynaecologists, and cervical screening service staff.

## Data Availability

Not applicable.

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
