# Peer review of "Stakeholder Perspectives of Australia’s National HPV Vaccination Program"

_vaccines, 2022, doi:10.3390/vaccines10111976_

Round 1
Reviewer 1 Report
Thank you for opportunity to review this paper. I have only a few minor comments to make, as the survey is well-conducted and most interesting insights seem to have been captured.
Perhaps the authors could be more surgical in their extraction of relevant details from the quotations provided.
The authors could also consider identifying any compelling results by a certain disaggregation that could be added as an additional figure.
Please consider mentioning the value of HPV vaccination for oral cancers (line 38).
Reviewer 2 Report
Summary:
Australia changed its’ recommended HPV vaccination schedule in 2018 from a 4-valent to 9-valent HPV vaccine and a 3-dose to 2-dose standard schedule. This manuscript records perspectives of the national vaccination program. The respondents were primarily people who worked for government/state/tribal agencies during the survey either a 4- or 6-month period in Australia (depending on whether data collection occurred October 2019-March 2020 (page 2) or November 2019-February 2020 (page 3). The survey results suggest the new two-dose HPV vaccination is accepted but that compliance rates have not changed.
Suggestions:
The Introduction could be shortened by removal (or shortening) of the first five sentences of the manuscript. These sentences relate to the biology of the HPV virus – this is NOT a biological project. To start the manuscript this way suggests to the reader that something about the biology of HPV will be investigated and/or discussed in the project. The Manuscript could start with “The higher burden of disease ….” On line 44 of the first page. Perhaps the biology of HPV can be placed at a more appropriate location in the manuscript.
The emphasis of the Introduction should not be on cervical cancer at the national level since only about half the population of most countries has a cervix and the HPV vaccine is recommended for both males and females. A multitude of ailments are associated with HPV infection and provide the basis for a broad national vaccine strategy. This approach might shorten the Introduction by a additional paragraphs.
The Results are presented in an organized way. However, the overview of the methods could be improved by a more specific indication of how the questionnaire used was ‘tailored’ for each of the stakeholder groups ahead of the interviews.
The Results sections contains observational comments and quotes from the respondents. These are presented without any indication of how often similar sentiments were observed. As a result, there is no way to determine whether or not the authors are cherry picking their presentation in regard to these conclusory statements. Removing the quoted items and replacing them with analyses of how often similar sentiments were obtained in the survey would help the reader understand the results of the survey in a quantitative way rather than a potentially biased sampling way. The repeated use of terms such as ‘many’, ‘more’, ‘several’, and ‘common’ are used in this regard without a quantitative explanation of how these terms are being used.
The relatively large proportion of ‘Don’t Know or N/A’ responses in Figure 1 and Figure 2 are basically ignored. Given that over 30% of respondents over multiple questions answered in these ways should be noted and discussed relative to what the authors say are the take-home points from their survey.
The Results section does not address the data showing that most of the respondents were middle age and older females. The preponderance of older female respondents should be at least mentioned given the historical content of HPV infections.
The Discussion needs to be shortened. The Discussion should focus on things related to the data obtained. There is too much in the Discussion that could have been written before the results of the survey were known.
The Discussion does not address the possibility that Australia has achieved ceiling HPV immunization rates in both males and females. This is especially true if the conclusion that vaccine hesitancy is not an important factor in HPV immunization in Australia.
Overall Impression
Can be shortened. Can be more focused. Suggest a deeper dive into the survey data to support some of the conclusory statements made.
